# Association between the Concentrations of Essential and Toxic Elements in Mid-Trimester Amniotic Fluid and Fetal Chromosomal Abnormalities in Pregnant Polish Women

**DOI:** 10.3390/diagnostics12040979

**Published:** 2022-04-13

**Authors:** Joanna Suliburska, Jakub Pankiewicz, Adam Sajnóg, Magdalena Paczkowska, Beata Nowakowska, Ewa Bakinowska, Danuta Barałkiewicz, Rafał Kocyłowski

**Affiliations:** 1Department of Human Nutrition and Dietetics, Poznan University of Life Sciences, Wojska Polskiego St. 31, 60-624 Poznan, Poland; 2PreMediCare New Med Medical Center, ul. Drużbickiego 13, 61-693 Poznan, Poland; biuro@new.med.pl (J.P.); rkocylow@gmail.com (R.K.); 3Department of Trace Analysis, Faculty of Chemistry, Adam Mickiewicz University, ul. Uniwersytetu Poznanskiego 6, 61-614 Poznan, Poland; adam.sajnog@amu.edu.pl (A.S.); danutaba@amu.edu.pl (D.B.); 4The Institute of Mother and Child, ul. Kasprzaka 17a, 01-211 Warsaw, Poland; magdalena.paczkowska@imid.med.pl (M.P.); beata.nowakowska@imid.med.pl (B.N.); 5Institute of Mathematics, Poznan University of Technology, ul. Piotrowo 3A, 60-965 Poznan, Poland; ewa.bakinowska@put.poznan.pl

**Keywords:** pregnancy, elements, amniotic fluid, chromosomal abnormalities

## Abstract

The present study aimed to investigate the relationship between the concentrations of essential and toxic elements present in the amniotic fluid (AF) and fetal chromosomal abnormalities in pregnant women. A total of 156 pregnant white Polish women aged between 20 and 43 years and screened to detect high risk for chromosomal defects in the first trimester were included in the study. AF samples were collected from these women during routine diagnostic and treatment procedures at mid-gestation (15–22 weeks of their pregnancies). The concentrations of various minerals in the AF were determined by inductively coupled plasma mass spectrometry. Genomic hybridization and cytogenetic karyotyping were performed to detect chromosomal aberrations in the fetuses. The genetic analysis revealed chromosomal aberrations in 19 fetuses (over 12% of all the evaluated women). The major abnormalities identified were trisomy 21 (N = 11), trisomy 18 (N = 2), and triploidy (N = 2). Fetuses with chromosomal abnormalities more frequently showed lower manganese concentration in the AF in the second trimester as compared to those with normal karyotype. A coincidence was observed between high iron levels in the AF and a higher risk of chromosomal abnormalities in the fetuses.

## 1. Introduction

Amniotic fluid (AF) is an important diagnostic material, and it is commonly tested to detect chromosomal abnormalities, fetal anomalies, and diseases. Fetal urine begins to appear in the AF from the second trimester of pregnancy. During the early development of the fetus, the AF comes in contact with the fetal skin, gastrointestinal tract, and lungs. Therefore, it can be assumed that the composition of the AF reflects that of fetal plasma [1]. AF composition is influenced by the mother’s nutrition, including essential nutrients, and by the exposure of the mother to toxic substances [2].

To date, only a few studies have investigated the relationship between mineral nutrition and the concentration of toxic elements in the mother’s body, including the AF, with the risk of congenital and chromosomal abnormalities in fetuses and newborns [3,4,5]. Because Down’s syndrome (DS) or trisomy 21 is one of the most common chromosomal abnormalities detected in humans, most of the results indicate an association between this condition and the concentration of mineral elements. A meta-analysis showed that people with DS have lower concentrations of calcium (Ca), selenium (Se), and zinc (Zn) in serum and higher concentrations of Zn and copper (Cu) in red blood cells as compared to healthy individuals [3]. Altered iron (Fe) metabolism and neuroinflammation have been observed in patients with DS as well as in those with Alzheimer’s disease [4]. Moreover, it has been found that the prevalence of anemia due to Fe deficiency is higher among children with DS (around 18%) than in the general population (around 3%) [5]. The clinical symptoms of chromosomal abnormalities such as trisomy 18 and trisomy 21 include neurological disorders such as epilepsy and brain dysfunctions, which may lead to cognitive decline [6,7]. Some studies have suggested that amyloid overexpression in the brain is the key factor that contributes to cognitive decline in people [8]. Extracellular accumulation of Zn and Cu in the amyloid and intracellular excess of Fe in neurons were reported to be associated with DS in humans, whereas in an animal study, the activity of the Ca ion channel was observed to be higher in a mouse model of DS than in wild-type mice [8]. Some environmental factors, such as heavy metals, may increase the risk of chromosomal aberrations in pregnancy [9,10]. Metals such as cadmium (Cd), aluminum (Al), arsenic (As), and lead (Pb) are neurotoxic, and early exposure to these metals may cause neurodevelopmental disorders [9]. In our previous study, we found that some essential and toxic elements were associated with congenital anatomical defects and growth disorders. Our results revealed that low concentrations of magnesium (Mg) and vanadium (V) and high concentrations of Al, Cd, and Pb in the AF were associated with an increased risk of nongenetic congenital disabilities [10].

The effects of essential and toxic elements on maternal health and fetal development have not been fully investigated, and the underlying mechanisms remain unclear. The results of recent research indicate that mineral elements may be more important than initially assumed in assessing the risk of chromosomal abnormalities in human fetuses [3,5]. Therefore, in this exploratory study, we investigated the coincidence between selected essential and toxic elements present in the AF and fetal chromosomal aberrations among white Polish pregnant women in the mid-trimester.

## 2. Materials and Methods

### 2.1. Study Design

This retrospective cross-sectional study aimed to determine the levels of mineral elements in the AF of pregnant women who were classified to have a high risk for chromosomal abnormalities based on the first-trimester prenatal screening tests. The AF was collected from the participants during a routine prenatal invasive procedure (amniocentesis following amniopuncture) at 15–22 weeks of gestation. The eligibility of the women participants to undergo amniopuncture was based on the first-trimester screening following the protocols of the Fetal Medicine Foundation (FMF).

The inclusion criteria of the study were as follows: (1) single viable intrauterine pregnancy, (2) second trimester, (3) high risk for fetal chromosomal abnormality, and (4) informed consent. The exclusion criteria were as follows: multiple pregnancies, lack of/withdrawal of the consent, insufficient AF sample, excessive blood contamination in the AF, maternal or intrauterine infection, maternal chronic illnesses, and receiving multidrug therapy.

Data on maternity and previous pregnancy, along with a history of illnesses, surgeries, drugs, and medications taken, were collected during a routine medical interview. The gestation week was determined based on the crown–rump length measurement obtained from the first-trimester obstetric ultrasound scan. AF samples were collected from all women, and genomic hybridization and cytogenetic karyotyping were performed. According to the results of chromosome analysis, the participants were divided into two groups: the control group (with normal fetal karyotype, C) and the abnormal group (with fetal chromosomal abnormalities, A).

A total of 156 pregnant Polish white women aged between 20 and 43 years and receiving routine obstetric care were included in the study. Participation in the study was voluntary, and all women signed an informed consent form and agreed to the secondary use of their samples.

The study design was approved by the Bioethics Commission at Poznan University of Medical Sciences (approval No. 492/21) in Poland. The study was conducted in accordance with the Declaration of Helsinki and registered with ClinicalTrials.gov (ID: NCT03598361).

### 2.2. Collection of Biospecimens

For genetic and elemental analyses, AF samples were obtained (5 mL) during a routine diagnostic amniocentesis procedure at 15–22 weeks of gestation. The sample collection was performed by transabdominal aspiration by using sterile 22–25G spinal and diagnostic puncture needles and plastic syringes free from organic and nonbiological contamination. Each fresh AF sample was centrifuged for 5 min at 1300 rpm. Next, 2 mL of the resulting supernatant was transferred to a new 2 mL tube and frozen at –80 °C for further analysis.

### 2.3. Measurement of Mineral Elements

AF samples were digested in the heating block (AccuBlock Digital, Labnet, Edison, NJ, USA) as follows: 500 µL of the AF sample was transferred to 15 mL polypropylene Falcon tubes, and 0.5 mL of 65% nitric acid was added. The mixture was vortexed for 10 s and heated for 60 min at 100 °C. After cooling, the digested samples were diluted with Milli-Q water (Direct-Q-3 UV, Merck, Darmstadt, Germany) to a volume of 10 mL. Certified reference materials (CRMs) were used to evaluate trueness and establish the traceability of the measurement result: Trace Elements Serum L-1 LOT 1309438 (Seronorm, Billingstad, Norway) and Trace Elements Serum L-2 LOT 1309416 (Seronorm). Because of the lack of CRM with a matching matrix, serum-based CRM was used for the analysis. The reference materials and procedural blanks were digested in a manner similar to AF samples, with 500 µL of Milli-Q water instead of the AF. Inductively coupled plasma mass spectrometry was performed using an Agilent 7700× system (Agilent, Santa Clara, CA, USA) to determine the concentration of mineral elements in the digested AF samples. The instrumental parameters were automatically optimized using the Tuning Solution (Agilent) and are provided in Table 1.

The parameters of the analytical method were estimated based on repeated measurements of blank solutions, calibration standards, and CRMs and are provided in Table 2.

The linearity of the calibration curve was calculated as the correlation coefficient (*R*), the value of which is greater than 0.999 for all analytes. The limit of detection (LOD) and limit of quantification (LOQ) were estimated based on the standard deviation (SD) of the repeated measurements of calibration blank samples by using the formulas: LOD = 3 SD and LOQ = 10 SD. The LOD values for Cd and Ca are 0.0073 and 210 µg/L, respectively. Precision was estimated based on three replicates of a CRM in a single analytical run and reflected the instrumental variability of the analytical signal. The precision values for Fe and Cd are 1.5% and 32.7%, respectively. Trueness was calculated as the recovery of the certified concentration by determining the analytes in the CRM. The recovery values ranged between 88% and 127% for Cr and Al, respectively. The results of the Student’s *t*-test also confirmed that there were no significant differences between the measured concentration ± SD and the certified concentration ± standard uncertainty.

### 2.4. Genomic Hybridization and Cytogenetic Karyotype Analysis

#### 2.4.1. DNA Isolation

For chromosomal microarray analysis, genomic DNA from fresh AF was extracted using a Sherlock kit (A&A Biotechnology, Gdansk, Poland) according to the manufacturer’s instructions.

#### 2.4.2. Array Design

Array comparative genomic hybridization (aCGH) was performed using 8x60K microarrays from Oxford Gene Technology (Begbroke, Oxfordshire, UK) (CytoSure ISCA, v3). The array contains 51,317-mer oligonucleotide probes covering the whole genome, with an average spatial resolution of 60 kb. Detailed description of the kit is available on the website (https://www.ogt.com/products/product-search/cytosure-constitutional-v3-and-v3-loh-arrays/, accessed on 7 March 2022).

#### 2.4.3. aCGH Analysis

The procedures of DNA denaturation, labeling, hybridization, and washing were performed in accordance with the manufacturer’s instructions. Genomic DNA was labeled using a CytoSure Labeling Kit (Oxford Gene Technology), without applying enzyme digestion. Hybridization was performed for 24–48 h at 65 °C in a rotator oven (Agilent). The arrays were washed using Agilent wash buffers 1 and 2 and scanned using a microarray scanner (Agilent Technologies), and signal intensities were then measured using Feature Extraction software (Agilent Technologies). All scanned images were quantified using Agilent Feature Extraction software (v10.0).

Data analysis was performed using CytoSure Interpret Software (Oxford Gene Technology) based on the reference genome (NCBI37/hg19).

#### 2.4.4. Cytogenetic Karyotyping

In most cases, a part of the AF submitted for aCGH analysis was also used for cytogenetic karyotyping.

Amniocytes were cultured in situ in vessels. Colony growth and mitotic activity were controlled every day after the 8th day of culture. Metaphase chromosomes were prepared following the standard procedures of GTG banding.

### 2.5. Statistical Analysis

All statistical analyses were performed in Statistica 13 for Windows and Rstudio software (R version 3.4.0; R Core Team, Vienna, Austria, 2017). The normal distribution of data was verified using the Shapiro–Wilk test. Differences between the groups were compared using the Mann–Whitney test. Depending on the values of mineral elements in the AF, the probability function of chromosomal abnormalities was determined. The symbol p0 indicates the probability of chromosomal abnormalities for different values of the analyzed elements. PCA was used to determine the correlations in the study population (based on the dataset). The level of statistical significance was set at *p* < 0.05 for all analyses.

The characteristics of the participants are shown in Table 3.

The study population comprised 156 pregnant women aged 20–43 years, who were in 15–22 weeks of gestation. Women in the fetal chromosomal aberration group were significantly older than those in the fetal normal karyotype group (*p* = 0.006), and the weeks of gestation were comparable in both groups (Table 3). The analysis of the biological material found in the AF showed that 49% were male fetuses and 51% were female fetuses, and the gender proportion was similar in both groups. Chromosomal abnormalities were identified in 19 fetuses (over 12% of all the evaluated women) and included trisomy 21 (N = 11), trisomy 18 (N = 2), triploidy (N = 2), and other aberrations (N = 4).

In this study, the concentrations of essential and toxic elements in the AF were determined. The content of these elements in the AF is presented in Figure 1 and Figure 2. The comparison of the analyzed mineral elements in the AF indicated a significant decrease in manganese (Mn) concentration in the group with chromosomal abnormalities as compared to that in the group with a normal karyotype (*p* = 0.0185) (Figure 1 and Figure 2).

The probability function of chromosomal aberrations in the fetuses based on the values of essential and toxic elements in the AF (Figure 3) indicated that the risk of chromosomal aberrations among fetuses significantly increased with the age of a pregnant woman (*p* = 0.013). Age over 40 years was associated with a more than 20% probability of fetal chromosomal aberrations. Moreover, an enhanced Fe concentration in the AF was markedly associated with the probability of having an abnormal fetal karyotype (*p* = 0.031). Fe concentration of approximately 1500 µg/L coincided with a >40% probability of fetal chromosomal aberrations. On the other hand, a decreased concentration of Mn in the AF increased the probability of the occurrence of aberrations (*p* = 0.098).

The main goal of principal component analysis (PCA) is to reduce the number of input data. In our work, we did not reduce the number of parameters tested. The PCA analysis was used to only visualize the correlation (dependence) between the examined parameters (Figure 4).

The vectors in Figure 4 represent the tested parameters: Mn, Al, nickel (Ni), chromium (Cr), Cd, Fe, Mg, Cu, Ca, Se, strontium (Sr), cobalt (Co), As, lithium (Li), and Age. The closer the two vectors are to each other, the more positive the correlation between the variables represented by these vectors, i.e., the examined parameters. As shown in Figure 4, the Cr and Ni vectors are very close to each other. This implies that as the Cr value increased, the Ni values also increased. The vectors Fe and Se are also very close to each other. This indicates that with the increase in Fe value, the value of Se also increases. In this study, two groups of strongly correlated parameters were noted. The first group included Mn, Al, Ni, Cr, Cd, and Age. The second group consisted of Fe, Mg, Cu, Ca, Se, Zn, and Sr. If two vectors are orthogonal, it indicates no correlation (no dependence) between the parameters. In the present study, the Ni vector is orthogonal to the Fe vector. This implies that an increase in the Ni value does not change the Fe value.

## 3. Discussion

In the present study, we analyzed the association between the concentration of essential and toxic elements in human AF with chromosomal abnormalities in fetuses of white Polish women in the second trimester of pregnancy. The results suggested that low concentrations of Mn and high concentrations of Fe coincided with a higher risk of fetal chromosomal abnormalities. To the best of our knowledge, our study is the first to demonstrate this relationship in Polish pregnant women.

The study showed that the probability of chromosomal aberrations was higher in fetuses with a low level of Mn in their AF. Mn plays a crucial role in maternal health and the early development of the fetus. Mn activates certain enzymes and acts through Mn-dependent metalloenzymes, such as manganese superoxide dismutase (MnSOD), which are needed for reproduction and the protection of the fetus against free radicals [11]. Some animal and human studies have shown that a low Mn level in pregnant women is associated with reduced reproductive function, impaired fetal growth, and poor perinatal outcomes [11]. Mn, when present in excess levels, however, can be a potent neurotoxin and may induce adverse neurological, reproductive, and respiratory effects [11]. The relationship between Mn status and chromosomal abnormalities is not fully known. The mitochondrial oxidative phosphorylation pathway generates a large number of free radicals, and MnSOD protects against these radicals while maintaining the oxidant/antioxidant balance [12]. A decrease in Mn level causes a reduction in the activity of MnSOD and disrupts this balance [12]. Oxidative stress is found to be increased in the AF of individuals with DS, and this might be linked to the low level of Mn in the AF [12,13].

Excess Fe can also lead to oxidative stress because Fe ions generate free radicals during the Fenton reaction. Hattori et al. [14] showed that the content of catalytic Fe(II) in the AF was significantly higher in women with abnormal pregnancies (also involving trisomy 21 and trisomy 18) than in those with normal pregnancies. The authors thus concluded that catalytic Fe in the AF may be a marker of abnormal pregnancy. Our study partly confirms this relationship and indicates that higher Fe levels are observed in the AF of fetuses with chromosomal abnormalities, mainly trisomy 21 (DS) and trisomy 18.

Other chromosomal trisomies are also associated with increased oxidative stress, inflammation, and nervous system disorders. However, most previous studies regarding the relationship between mineral levels and chromosomal abnormalities have focused on DS [15]. Significantly higher ferritin levels and lower transferrin and total iron-binding capacity (TIBC) levels were found in adults with DS (18–35 years) than in healthy individuals [15].

Increased oxidative stress observed in individuals with DS may begin in utero and play a crucial role in phenotypic traits [16]. DS and other chromosomal aneuploidies are characterized by metabolic diseases such as obesity, insulin resistance, and lipid disorders with greater involvement of oxidative stress in their etiopathogenesis [7,16]. Therefore, based on the obtained results, we can assume that a high level of oxidative stress associated with high Fe and low Mn concentrations in the AF during early fetal development may contribute to increased oxidative stress, brain disorders, and other metabolic disorders in the later life of individuals with aneuploidies. A treatment strategy that allows lowering oxidative stress by regulating the concentrations of selected mineral elements in the AF may potentially protect against the disorders observed in individuals with chromosomal aberrations.

The occurrence of chromosomal aberrations with a low Mn concentration and a high Fe concentration in the AF may also be associated with the interaction between Fe and Mn in terms of competing transport pathways. Divalent metal transporter 1 (DMT1) is the primary nonheme Fe transporter in the intestine that transports Fe and other elements, including Mn [11]. DMT1 is found not only in the intestine but also in other tissues, including the placenta, and the transport of an excessive amount of Fe through this transporter reduces the transport of Mn in tissues [11].

The concentration of mineral elements in the AF may be influenced by endogenous and exogenous factors. The correlations observed between the levels of Fe and Cu and between the levels of Ca and Mg indicate the transport and metabolic relationships between these elements [17]. Systemic Cu deficiency results in cellular Fe deficiency [17]. Cu and Fe participate in single-electron transfer reactions, and because of this functionality, they may generate free radicals and increase oxidative stress in the body [17]. As mentioned above, an association was found between excess Fe and neurodegenerative diseases, including DS [14]. Our present study showed slightly higher Fe levels in the AF in pregnancy with chromosomal disorders. In a mouse model of DS, Cu accumulation in the brain increased oxidative stress [18]. A change in Cu homeostasis in the brain is also suggestive of Alzheimer’s disease [8]. In the human body, the metabolism of Mg and Ca is closely related [19]. The intestinal absorption and renal excretion of these two elements are also related [19]. A positive correlation observed between toxic elements, e.g., Al and Ni, in our study may be the consequence of exposure of pregnant women to environmental pollution, which was also confirmed in other studies [20]. The correlation index values are higher for AF with chromosomal abnormalities, which may be due to the exposure of women in this group to different toxic elements during pregnancy. Toxic elements such as Cd, Ni, and Al are known to be associated with chronic morbidity [20,21]. McLachlan et al. [22] showed that Al accumulation in the brains of patients with Alzheimer’s disease and DS might contribute to the neuropathology of symptoms of these diseases.

Advanced age in women is associated with increased risks of infertility and aneuploidy in the offspring, mostly DS (trisomy 21), Edwards’ syndrome (trisomy 18), and Turner’s syndrome (monosomy X) [23]. Oxidative stress increases with human age and can alter gene expression; it may also lead to mitochondrial dysfunction in the cells of ovarian follicles, thus disturbing the activation of MnSOD [23]. Chromosome segregation may be erroneous during meiosis in aged oocytes, which can result in aneuploidy and lower the quality of oocytes [23,24].

The present study has some limitations. First, the study group consisted of a relatively small number of women with fetal chromosomal abnormalities and was also heterogeneous in terms of different fetal chromosomal abnormalities (almost 60% of the cases were DS). Furthermore, we did not determine maternal nutritional and environmental factors that may have affected the concentration of mineral elements in the AF. We also did not consider other maternal parameters such as smoking, body mass index, education, and socioeconomic status, which may influence the concentration of mineral elements in the AF. Additionally, we did not analyze any genetic anomalies in the pregnant women, which might extend our discussion regarding the obtained results. Moreover, some studies have shown that the concentration of Mn in mothers decreases with age. It is thus unclear whether low Mn levels in the AF are associated with chromosomal abnormalities in the fetus. However, our study did not show a significant relationship between the concentration of Mn and maternal age. Further studies with a large sample size are needed in the future to confirm this relationship.

## 4. Conclusions

A lower Mn concentration in the AF in the second trimester of pregnancy is more frequently associated with chromosomal abnormalities in human fetuses as compared to that in fetuses without chromosomal aberrations. A high Fe concentration in the AF also corresponds to a higher incidence of chromosomal abnormalities in fetuses. Further prospective studies with a larger population are required to derive an appropriate conclusion.

## Figures and Tables

**Figure 1 diagnostics-12-00979-f001:**
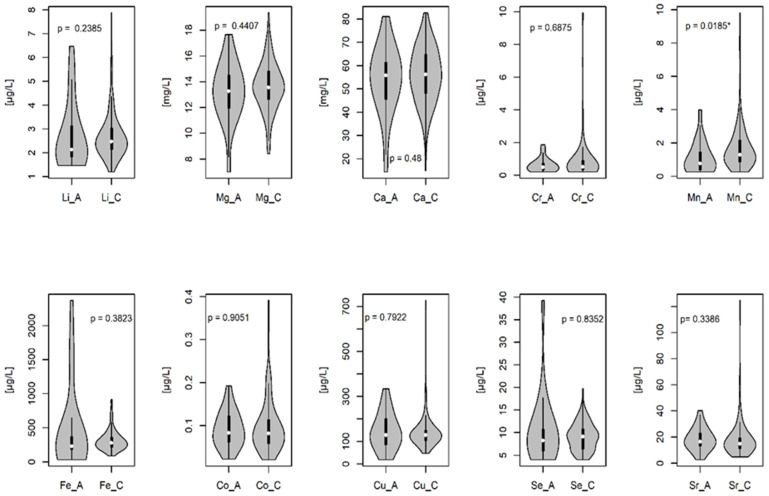
The concentration of essential elements in the amniotic fluid. C-group with normal fetal karyotype; A-group with fetal chromosomal abnormalities. The lower part of the violin chart is the minimum mineral concentration; the upper part of the violin chart is the maximum mineral concentration. The lower range of the black box is the lower quartile Q1; the upper range of the black box is the upper quartile Q3. The white dot in the black box is the median value; *p*-significance level. * statistically significant differences at the level of α = 0.05.

**Figure 2 diagnostics-12-00979-f002:**
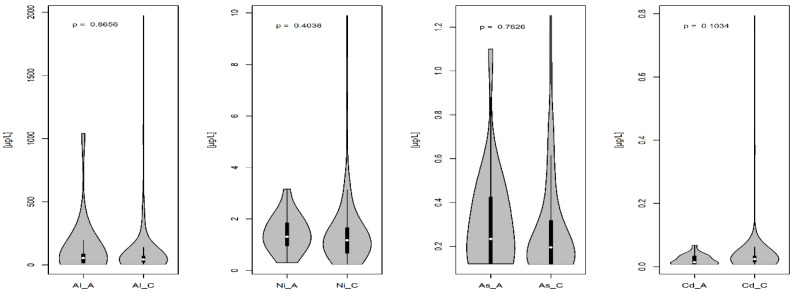
The concentration of toxic elements in the amniotic fluid. C-group with normal fetal karyotype; A-group with fetal chromosomal abnormalities. The lower part of the violin chart is the minimum mineral concentration; the upper part of the violin chart is the maximum mineral concentration. The lower range of the black box is the lower quartile Q1; the upper range of the black box is the upper quartile Q3. The white dot in the black box is the median; *p*-significance level.

**Figure 3 diagnostics-12-00979-f003:**
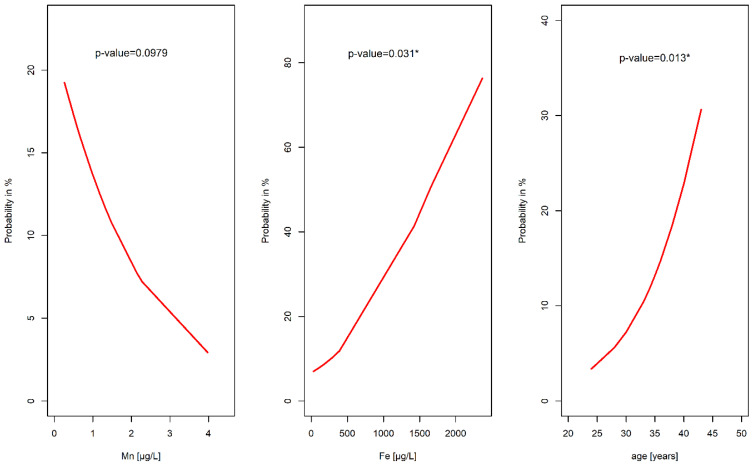
The probability of fetal chromosomal abnormalities in relationship with the concentrations of Mn and Fe in the amniotic fluid and with the maternal age; * statistically significant relationships at the level of α = 0.05.

**Figure 4 diagnostics-12-00979-f004:**
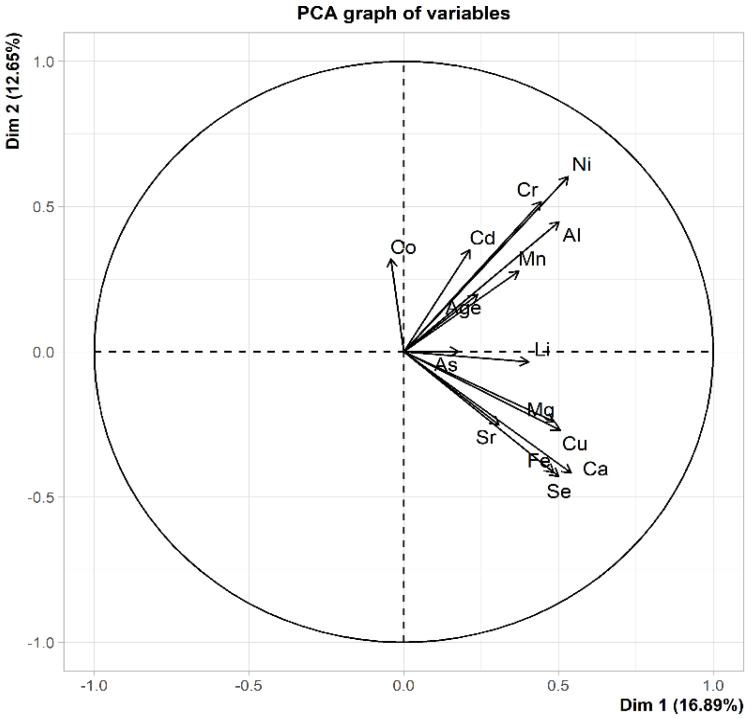
Correlation chart showing the relationships between the examined parameters: Mn, Al, Ni, Cr, Cd, Fe, Mg, Cu, Ca, Se Sr, Co, As, Li, and Age; the relationships are presented in the system of the first two principal components Dim 1 and Dim 2. These two components are a combination of all the tested parameters.

**Table 1 diagnostics-12-00979-t001:** Operating conditions of ICP-MS.

Parameter	Setting
Instrument	Agilent 7700×
Cones	Ni
Nebulizer gas flow (L min^−1^)	0.95
Auxiliary gas flow (L min^−1^)	0.90
Plasma gas flow (L min^−1^)	15
RF Power (W)	1550
Peristaltic pump speed (rps)	0.10
He flow (mL min^−1^)	4.2
Integration time (s)/Monitored isotopes	0.1/^7^Li, ^24^Mg, ^27^Al, ^43^Ca, ^52^Cr, ^55^Mn, ^56^Fe, ^59^Co, ^60^Ni, ^63^Cu, ^88^Sr 1.0/^75^As, ^82^Se, ^111^Cd
Replicates	3
Sweeps per replicate	100

The calibration standards were prepared from multielement calibration standard 3 (PerkinElmer, Waltham, MA, USA) and single element calibration standards of Mg and Ca (PerkinElmer) in the following concentrations: 0.1, 0.5, 1, 5, 10, 25, and 50 mg/L for Mg and Ca, and 0.1, 0.5, 1, 5, 10, 25, and 50 µg/L for other analytes. The calibration blank sample signals were subtracted from the standard solution signals.

**Table 2 diagnostics-12-00979-t002:** Results of the validation of element analysis—parameters of the analytical method.

	Li	Mg	Al	Ca	Cr	Mn	Fe	Co	Ni	Cu	As	Se	Sr	Cd
Linearity *R*	0.9994	0.9992	0.9993	0.9991	0.9998	0.9998	0.9998	0.9999	0.9999	0.9995	0.9999	0.9998	0.9997	0.9998
LOD [µg/L]	0.22	11	4.9	210	0.20	0.26	0.47	0.021	0.24	0.83	0.12	4.0	0.16	0.0073
LOQ [µg/L]	0.72	37	16	690	0.65	0.85	1.6	0.071	0.80	2.8	0.39	13	0.52	0.024
Precision [%]	4.0	2.1	5.1	3.7	9.5	18.9	1.5	24.2	9.9	2.5	7.4	30.6	7.0	32.7
Trueness [%]	99	99	127	104	88	114	89	99	109	105	93	125	103	115

LOD—the limit of detection; LOQ—the limit of quantification.

**Table 3 diagnostics-12-00979-t003:** Characteristic of the groups (mean ± SD/median/min-max).

Parameters	Fetal Normal Karyotype	Fetal Chromosomal Aberrations
**N**	137	19
**Age (years)**	31.72 ± 5.42/32/20–43 ^#^	35.21 ± 4.50/36/24–43 ^#^
**Week of gestation**	18.13 ± 2.91/17/15–22	18.10 ± 2.71/17.5/15–22
**Fetal gender (F/M)**	70/67	10/9
**Chromosomal aberration (N):**	0	19
**Trisomy +18**	0	2
**Trisomy +21**	0	11
**Triploidy**	0	2
**mos 47,XY,+5.nuc ish 5q21(EGR-1x3)[24/64]**	0	2
**46,XX,del(4)(p16.1)**	0	2

^#^—statistically significant; Mann–Whitney test; *p* = 0.006.

## Data Availability

The data used to support the findings of this study can be made available by the corresponding author upon request.

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
