# Peer review of "Association between the Concentrations of Essential and Toxic Elements in Mid-Trimester Amniotic Fluid and Fetal Chromosomal Abnormalities in Pregnant Polish Women"

_diagnostics, 2022, doi:10.3390/diagnostics12040979_

Round 1
Reviewer 1 Report
In this study the authors aim to investigate the link between essential and toxic elements present in the amniotic fluid (AF) and chromosomal abnormalities in pregnant women at midgestation. Their findings indicated that human fetuses with chromosomal abnormalities were more frequently characterized by lower manganese concentration in AF in the second trimester of pregnancy compared to healthy ones. High iron levels in AF also correspond with a higher risk of chromosomal abnormalities in human fetuses.
Introduction
-Lane 51: add the space before [3]
-Lane 83: Table 3 must be replaced with Table 1
Lane 103: Remove capital letter from Indicated
-About this sentence: In our previous study, we found that some essential and toxic elements were associated with congenital anatomical defects and growth disorders. Our results revealed that low concentrations of magnesium and vanadium and high concentrations of aluminum, cadmium, and lead in AF were associated with an increased risk of nongenetic birth defects [10]
The objective of this study was to determine the concentration and the reference ranges of essential and toxic elements in amniotic fluid (AF) and maternal serum (MS) at birth. I did not find the relationship between the essential and toxic elements and congenital anatomical defects and growth disorders.
Results
-In the paragraph below:
“The probability function of chromosomal anomalies in fetuses analyzed based on the 102 values of elements in AF (Fig. 3) Indicated that the risk of chromosomal abnormalities 103 increased with the age of a pregnant woman. Age over 40 years was associated with a 104 more than 20% probability of chromosomal aberrations. Moreover, an enhanced iron con- 105 centration in AF increased the probability of an abnormal karyotype. An iron level of 106 about 1500 µg/L was associated with an over 40% probability of chromosomal anomalies. 107 For manganese, a decreased concentration in AF increased the probability of the occurrence of anomalies (p=0.098)”
I suggest to add the sentences about the statistical significance referring to p-value
Discussion
Regarding this sentence: “
The results suggested that low concentrations of manganese and high concentrations of iron were linked with a higher risk of chromosomal abnormalities (mainly trisomy 21, DS).
Do the authors base this sentence on reliable data or simply on the high number of Down Syndrome (60%) among the group with chromosomal abnormalities?
Author Response
Dear Editor and Reviewer,
We are very grateful for your comments on our manuscript. We have revised the manuscript in accordance with your advice, and carefully proof-read it to minimize any errors.
Reviewer 1
Introduction
-Lane 51: add the space before [3]
-Lane 83: Table 3 must be replaced with Table 1
Lane 103: Remove capital letter from Indicated
Response: It has been changed.
About this sentence: In our previous study, we found that some essential and toxic elements were associated with congenital anatomical defects and growth disorders. Our results revealed that low concentrations of magnesium and vanadium and high concentrations of aluminium, cadmium, and lead in AF were associated with an increased risk of nongenetic birth defects [10]
The objective of this study was to determine the concentration and the reference ranges of essential and toxic elements in amniotic fluid (AF) and maternal serum (MS) at birth. I did not find the relationship between the essential and toxic elements and congenital anatomical defects and growth disorders.
Response: We would like to note that we conducted other studies where we found relationships between minerals in AF and maternal serum with other disorders in fetuses.
Results
-In the paragraph below:
“The probability function of chromosomal anomalies in fetuses analyzed based on the 102 values of elements in AF (Fig. 3) Indicated that the risk of chromosomal abnormalities 103 increased with the age of a pregnant woman. Age over 40 years was associated with a 104 more than 20% probability of chromosomal aberrations. Moreover, an enhanced iron con- 105 centration in AF increased the probability of an abnormal karyotype. An iron level of 106 about 1500 µg/L was associated with an over 40% probability of chromosomal anomalies. 107 For manganese, a decreased concentration in AF increased the probability of the occurrence of anomalies (p=0.098)”
I suggest to add the sentences about the statistical significance referring to p-value
Response: It has been added.
Discussion
Regarding this sentence: “The results suggested that low concentrations of manganese and high concentrations of iron were linked with a higher risk of chromosomal abnormalities (mainly trisomy 21, DS).
Do the authors base this sentence on reliable data or simply on the high number of Down Syndrome (60%) among the group with chromosomal abnormalities?
Response: Yes, we wrote this sentence because in our study group were more cases with DS. It has been changed.
Reviewer 2 Report
This is a well written and interesting article, very few reports were focused on the relationship between toxic elements including minerals and the occurrence of aneuploidy. It is still unclear such association is a coincidence or there is a causal relationship. However, such pioneering work deserves publication even if it is later proved unrelated. The Discussion has addressed the possible explanations regarding such association.
Author Response
Dear Editor and Reviewer,
We are very grateful for your comments on our manuscript. We have revised the manuscript in accordance with your advice, and carefully proof-read it to minimize any errors.
Reviewer 2
This is a well written and interesting article, very few reports were focused on the relationship between toxic elements including minerals and the occurrence of aneuploidy. It is still unclear such association is a coincidence or there is a causal relationship. However, such pioneering work deserves publication even if it is later proved unrelated. The Discussion has addressed the possible explanations regarding such association.
Response: Thank you for your comments.
Reviewer 3 Report
The manuscript submitted by Suliburska et al. concerns an interesting study of the mid trimester amniotic fluid elements and possible associations with the chromosomal abnormalities found in pregnant women.
First of all, the manuscript needs English correction by professional native speaker. It should be also followed by precise editorial improvement to avoid unnecessary/lack of i.e. spacers.
Metals should be written from the very beginning also with their symbols, i.e. manganese (Mn).
line 28, 102: not 'anomalies', rather 'aberrations'
l29, 85: 'total population' - in what sense? of all evaluated women within the study or literature or WHO data of general population?
l32, 229: 'healthy ones': ? non-pregnant? general population? please, clarify
l30-33, 105-109, 133-137, 179-181: it should be rewritten, because it is too hard (and not evidenced) statement that lower concentration of manganese or high iron levels is correlated to chromosomal abnormalities. It should be rather written as 'coincidental' or 'observed'; the same for the aim of the study (l72-75)
l37/38: 'AF is used in procedures' - no, AF is important for the pregnancy; it has to be rewritten in some way as: AF probing or sampling is taken for...
l46, 49, 57, 61, 72, 138, 140, 142, 144, 148, 152, 164, 165, 170, 178, 181, 183, 189 (2x), 197, 208: lack of references
l50: in blood or blood serum?
l51/52: and what about general population or in other syndromes/diseases?
l54/57: any linkage between those two sentences?
l61: what factors? please, add some examples
l62: early exposure on what? + more refs
Table 1:
legend: characteristics of the groups: please, add more information about what 'gender, chromosome abnormalities mean', etc:
what 'Gender' means? gender of what? It is unclear, because you stated n=156 women (normal karyo n=137 + abnormal n=19)
'chromosome abnormalities' in what? AF? it has to be clearly written
l79: 'statistically significant' rather than 'significant differences'; ab suggests two various values but here is only one p=0.006, please, correct this
which group is the control one? it is totally unclear/lack of description
l81, 87 and any other statements with 'significant' value: p value?
l83: Table 3? 'karyotype analysis' of what?
l86: n for 'other aberrations?
Fig 1 & Fig 2 can be merged into one or some longer description has to be added to the text to explain, why there are two separates (essential vs. toxic)
l113: PCA? please, explain; 'two principal components' - which one?
l117: 'whole study population': from literature/WHO/own data? which one?
l118: 'two groups' of what?
l113-128: unclear; please rewrite
l147: explain 'MnSOD'
l149, 160-162: totally unclear sentences
l157-160: too speculative; please, rewrite
l198-202: have you check the possible exposure for your evaluated group? there is no such data in 5.1. clarify/rewrite
l208: mechanism of what? +refs
l210: in what way? +refs
l210-212: rewrite, because it means that oocytes are being aged during meiosis, not the woman; add more refs here
5.1. please reorder: 244-255 after 239; then 256-259 after 243
l262: according to what protocol?
Table 3: legend unclear - validation of what?
5.4.2. short description recommended
l328: in situ in coursive.
Author Response
Dear Editor and Reviewer,
We are very grateful for your comments on our manuscript. We have revised the manuscript in accordance with your advice, and carefully proof-read it to minimize any errors.
Reviewer 3
First of all, the manuscript needs English correction by professional native speaker. It should be also followed by precise editorial improvement to avoid unnecessary/lack of i.e. spacers.
Response: The text has been edited by native speaker.
Metals should be written from the very beginning also with their symbols, i.e. manganese (Mn).
Response: Symbols of metals have been added.
line 28, 102: not 'anomalies', rather 'aberrations'
Response: We have changed that.
l29, 85: 'total population' - in what sense? of all evaluated women within the study or literature or WHO data of general population?
Response: It means: All evaluated women. The sentence has been changed.
l32, 229: 'healthy ones': ? non-pregnant? general population? please, clarify
Response: It means: group with normal karyotype. It has been changed.
l30-33, 105-109, 133-137, 179-181: it should be rewritten, because it is too hard (and not evidenced) statement that lower concentration of manganese or high iron levels is correlated to chromosomal abnormalities. It should be rather written as 'coincidental' or 'observed'; the same for the aim of the study (l72-75).
Response: It has been changed.
l37/38: 'AF is used in procedures' - no, AF is important for the pregnancy; it has to be rewritten in some way as: AF probing or sampling is taken for...
Response: The sentence has been rewritten.
l46, 49, 57, 61, 72, 138, 140, 142, 144, 148, 152, 164, 165, 170, 178, 181, 183, 189 (2x), 197, 208: lack of references
Response: The references have been added accordingly.
l50: in blood or blood serum?
Response: It has been clarified.
l51/52: and what about general population or in other syndromes/diseases?
Response: It has been clarified.
l54/57: any linkage between those two sentences?
Response: It has been added.
l61: what factors? please, add some examples
Response: It has been added.
l62: early exposure on what?
Response: It has been added.
Table 1:
legend: characteristics of the groups: please, add more information about what 'gender, chromosome abnormalities mean', etc:
what 'Gender' means? gender of what? It is unclear, because you stated n=156 women (normal karyo n=137 + abnormal n=19)
'chromosome abnormalities' in what? AF? it has to be clearly written
Response: It has been explained and added in Table 1.
l79: 'statistically significant' rather than 'significant differences'; ab suggests two various values but here is only one p=0.006, please, correct this
Response: It has been changed and corrected.
which group is the control one? it is totally unclear/lack of description
Response: The sentences have been rewritten.
l81, 87 and any other statements with 'significant' value: p value?
Response: It has been added.
l83: Table 3? 'karyotype analysis' of what?
Response: It has been added.
l86: n for 'other aberrations?
Response: It has been added.
Fig 1 & Fig 2 can be merged into one or some longer description has to be added to the text to explain, why there are two separates (essential vs. toxic)
Response: It has been explained in the text. We would like to emphasize that we analysed not only essential but also toxic elements as presented in our previous articles.
l113: PCA? please, explain; 'two principal components' - which one?
l117: 'whole study population': from literature/WHO/own data? which one?
l118: 'two groups' of what?
l113-128: unclear; please rewrite
Response: Entire part related to PCA has been rewritten.
l147: explain 'MnSOD'
Response: It has been explained.
l149, 160-162: totally unclear sentences
Response: It has been rewritten.
l157-160: too speculative; please, rewrite
Response: We agree with the Reviewer and we decided to remove these sentences.
l198-202: have you check the possible exposure for your evaluated group? there is no such data in 5.1. clarify/rewrite
Response: We did not analyse such exposure, we refer to other studies.
l208: mechanism of what? +refs
Response: It has been changed.
l210: in what way? +refs
Response: It has been changed.
l210-212: rewrite, because it means that oocytes are being aged during meiosis, not the woman; add more refs here
Response: It has been changed.
5.1. please reorder: 244-255 after 239; then 256-259 after 243
Response: It has been changed.
l262: according to what protocol?
Response: It should be ‘study design’ instead of ‘protocol’. It has been changed.
Table 3: legend unclear - validation of what?
Response: It has been explained.
5.4.2. short description recommended
Response: It has been added.
l328: in situ in coursive.
Response: It has been changed.
Round 2
Reviewer 3 Report
The manuscript by Suliburska et al. has been improved properly. However, it still needs English correction by professional native speaker. At this moment it looks that only remarks/corrections suggested by the Reviewer have been done, without the full English correction. Additionally, some minor points (mostly language ones) need to be corrected:
line 82: AF definition - I am not sure if it is needed here, because there are no AF data in the Table 1.
line 84: 'women in the group' rather 'from the group'
line 86: 'The karyotype analysis of the AF' - rather 'analysis of the biological material found in AF'
line 91: non grammatically written
line 109: what 'elements'?
lines 112/113: again: Fe is not causative for karyotype; please, rewrite
line 304: maybe it would be better to put the whole Table 3 into the next page? Also, below the Table 3 there are no footnotes concerning terms included in the table (LOD, LOQ).
Author Response
Dear Editor and Reviewers,
We are very grateful for your comments on our manuscript. We have revised the manuscript in accordance with your advice and carefully proofread it to minimize any errors.
The manuscript by Suliburska et al. has been improved properly. However, it still needs English correction by professional native speaker. At this moment it looks that only remarks/corrections suggested by the Reviewer have been done, without the full English correction.
Response: the text has been corrected by Translmed, we attached the edited text ver 1 and ver 2.
T|P|G
38 Hawthorne Dr, Unit E106
Bedford, NH, 03110, U.S.A.
Phone: +1 413.800.1204
E-mail: translmed@translmed.com
Website: https://translmed.com/
Additionally, some minor points (mostly language ones) need to be corrected:
line 82: AF definition - I am not sure if it is needed here, because there are no AF data in the Table 1.
line 84: 'women in the group' rather 'from the group'
line 86: 'The karyotype analysis of the AF' - rather 'analysis of the biological material found in AF'
line 91: non grammatically written
line 109: what 'elements'?
lines 112/113: again: Fe is not causative for karyotype; please, rewrite
line 304: maybe it would be better to put the whole Table 3 into the next page? Also, below the Table 3 there are no footnotes concerning terms included in the table (LOD, LOQ).
Response: All points have been revised and changed.
